# Through the eyes of the beholder – a study on zero-shot digitization of lab measurements

**Anonymous**                    ANONYMOUS@EMAIL.COM *Anonymous Institution*

## Abstract

The automatic tracking of human actions in scientific pipelines can enormously improve their reproducibility and reliability, ensuring accountability, quality control, traceability, and replicability of protocols. Self-recording is a non-intrusive strategy to create a visual diary, and egocentric video recordings store detailed information about the actions performed without influencing their ordinary course. However, raw videos generate massive amounts of unlabeled data that are challenging to index and use for information retrieval. In this paper, we study how gaze information can be beneficial to analyzing egocentric video recordings and automatically extracting accurate measurements about first-person procedures and manual operations. We propose a novel approach that uses gaze tracking to perform a fine-grained segmentation of the raw video content at the temporal and spatial levels. Based on gaze-driven segmentation, we then devise a methodology to extract precise quantitative information about two types of human actions: the measurement of a liquid volume and the weighting of an object on the scale. Both actions are examples of repetitive measurements performed in a laboratory, requiring high reproducibility. Results show that gaze has apparent benefits in terms of temporal segmentation and computational costs of information extraction. With this, we wish to open the discussion on gaze-based prompting to obtain real-world measurements from unlabeled egocentric video recordings, leveraging the recent advances in foundation models for image segmentation. Our proposed examples show how the synergies between gaze estimation and computer vision can facilitate the annotation of precise information, and we foresee that they will facilitate the natural interaction of human or robotic operators in scientific environments.

**Keywords:** Gaze, computer vision, lab automation

## 1. Introduction

With the large availability of storage facilities and smart edge devices that can capture raw data, such as smartphones, smartwatches, and smart googles, technologies that can automate action tracking and data collection are becoming increasingly likely to benefit several human activities. Importantly, high-risk pipelines where tasks can hardly be interrupted to annotate details, could benefit from automated tracking. The details of intraoperative surgical actions, for example, are rarely reported because of the surgeon's impossibility of stopping the procedure, and this is known to cause unpredictability in the outcomes (Kiyasseh et al., 2023). Moreover, in industrial settings where users perform tedious or repetitive tasks, the automated annotation of actions and specific parameters can improve the quality control, productivity and safety of the workflows. However, despite the current capability of collecting large scale collections of video recordings, the efficient indexing and retrieval of information from such large data collections is becoming increasingly challenging. We argue that gaze-prompting can be useful in all of those applications where precise values need to be annotated constantly but are easily overlooked because of the impossibility to stop the

action, safety measures, the need of immediacy or human inattention. Similarly, with low reproducibility being one of the problems in current research (Gonçalves and Musen, 2019; Open Science Collaboration, 2015), we believe that tracking actions during experimentation can lead to improved reproducibility of scientific findings.

Egocentric video recording is an excellent means to acquire raw data at large scale, providing precise information about the operations performed without interfering with the operator. However, the information of interest about the type of task being performed and the parameters involved represent an extremely small fraction of the recording data, and storing constant recording streams brings additional challenges such as storage limitations or the human effort required to review videoclips (Chang et al., 1999). Gaze estimation devices are becoming increasingly popular and several can now perform video recording and gaze estimation in a non intrusive, easy-to-use way. When combined with egocentric video, gaze estimation adds a stream of information that can be used to annotate the data stream with relevant information. Instead of adding complexity to the task, gaze can provide a method to identify information of interest within the video clip and a specific frame. Additionally, computer vision and visual foundation models can leverage these signals to not only describe actions with categorical values (e.g., which action is being performed) but also estimating numerical features of the action (e.g., for how long, how much, how far).

In this paper, we evaluate the use gaze estimation in egocentric video recordings to infer the associated numerical values of two human actions in a laboratory setting: volume estimation of a liquid measurement and mass estimation of solids on a scale. Both use cases represent tasks that are repeatedly performed in an environment where reproducibility is highly desired, but often missed due to the safety measures or the intrinsic difficulty of correctly annotating and keeping track of multiple values. These examples show how the synergies between gaze estimation and computer vision can reduce the burden in some contexts. In our study, we investigate the application of gaze estimation to prompt dual segmentation: temporal segmentation to identify the frame(s) of interest and spatial segmentation to identify the object that the user is looking at. A task specific logic is used to either estimate the volume in a liquid container or the mass of solids through digit recognition on a digital scale.

The rest of the paper is organized as follows: Section 2 describes the related work on gaze estimation for egocentric videos and still images, Section 3 materials (capture devices) used for our setup as well as the frame selection and segmentation strategies used, and the task-specific logic for each of the final tasks. Results of the experiments described in Section 4 are presented in Section 5 and discussed in Section 6. Conclusion is left for Section 7.

## 2. Related Work

The increasing capacity of recording and storing large amounts of visual data from a wearable capture device is matched by the increasing complexity of indexing and exploiting such amounts of data. The segmentation of crucial information from videos can be split in two independent processes along the spatial and temporal axes.

Temporal segmentation of videos can be performed with a variety of methods. Simple methods like pixel-wise differences (Kikukawa and Kawafuchi, 1992) or variations of

it (Zhang et al., 1993; Nagasaka and Tanaka, 1992), or color histograms in various color spaces (Gargi et al., 1995) are often limited by fast camera movements, that tend to produce a sudden change in the histogram or pixel difference features.

Some authors like Arman et al. (1993) proposed to benefit from the compression algorithms used for the video, which systematically measure and store only differences between frames. The use of *keyframes* (i.e., full storage of a frame without reference to previous or posterior frames) is optimized to reduce the size of the compressed files, and tend to be aligned with scene changes. Although cheaper to compute than pixel-wise differences (because the by-product of compression is re-used), these methods might also identify a fast camera movement as a change of scene, and therefore a temporal segment.

Most recent methods like the ones from Li et al. (2023b); Xin et al. (2023) have moved towards deep neural networks to learn what a temporal segment is without explicitly modeling what are the decisive features that drive the decision. This comes, however, at the cost of data and annotation, which explains why Fayyaz and Gall (2020) have introduced weak supervision in the training.

Other methods have integrated an additional source of information that is easier to process to help the temporal segmentation of the video. Such methods have often used audio signals or even speech recognition, but direct supervision of audio segmentation using video-based segmentation can be challenging or even counter-productive, also because of the misalignment of signals, as explained by Wu and Yang (2021). In egocentric videos, the situation can become even more challenging, since fast camera movements are frequent in first person video recording. In addition, audio signals can be misleading or simply too noisy to rely on them for temporal segmentation. We propose to use the gaze as an additional signal that is synchronously obtained with the video, that is extremely lightweight and can provide additional information for spatial segmentation.

Isolating spatial region of interest in static images has been of interest in the computer vision community for multiple purposes, from understanding scenes and recognizing actions (Cho and Kang, 2021), to facilitating the use of interactive technologies and boosting the efficiency of technical operators (Wang et al., 2023). The wide success of prompted segmentation algorithms such as the Segment Anything Model (Kirillov et al., 2023) (SAM) suggests that the combination of gaze tracking prompts and segmentation could drastically speed up the research in this direction, facilitating object detection, segmentation and recognition. The work by (Wang et al., 2023) pioneered this direction, using gaze prompts to segment organs in radiology scans. However, this approach mainly focused on static images in the specific domain of 3D computerized tomography, and there is still limited literature on how to adapt the existing works to object segmentation in videos and images in the wild. Other approaches focused on the detection problem rather than segmentation Santella et al. (2006); Cho and Kang (2021); Weber et al. (2022), facing the challenges that emerge when objects are in cluttered environments (Photchara et al., 2021) or located at multiple distances from the camera (Cho and Kang, 2021). Weber et al. (2022) proposed a method to detect unknown objects based only on the gaze information and its combination with a classification algorithm that retains only the gaze points falling on an object. The solution in Cho and Kang (2021) estimates the 3D gaze coordinates to combine the spatial information with the depth information to identify objects at multiple distances and scales. In gaze tracing (Photchara et al., 2021), the user is asked to follow a specific gazing and

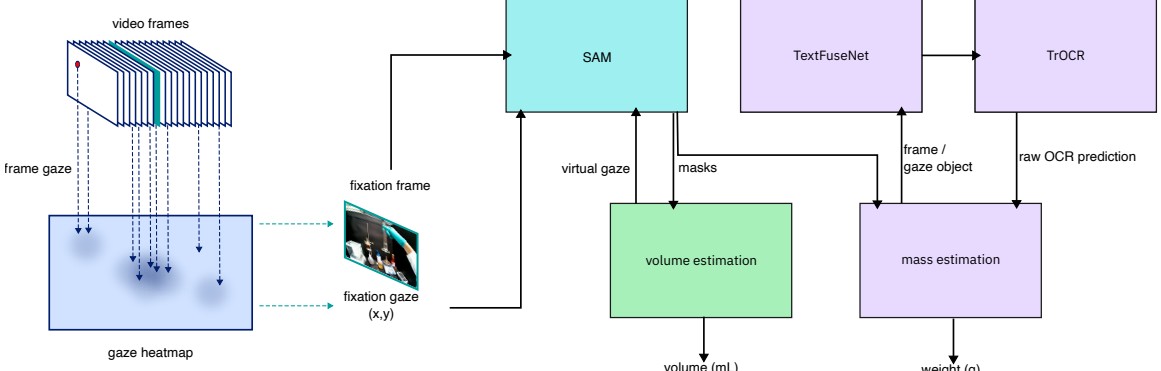

Figure 1: Schematic overview of the volume and mass estimation pipelines. The gaze signal is used to identify the fixation frame from within the video. The fixation frame and the gaze are used together for prompting segmentation or identifying objects of interest in panoptic segmentations. These segmentation masks are then used for volume estimation based on the mask size or to provide an input to the optical character recognition model.

blinking protocol to clarify the object of interest, that is then segmented automatically by pixel clustering. Finally, Shi et al. (2017) proposed a probabilistic approach to de-noise gaze coordinates and attribute pixel regions to the same segmentation outcome.

In contrast with these methods, we study approaches that exploit the potential of recent developments in segmentation algorithms such as SAM (Kirillov et al., 2023) to demonstrate that gaze prompting can be a valuable means to solve complicated tasks such as the segmentation of small objects such as graduated cylinders in the cluttered environment of a chemistry lab and the consequent estimation of liquid volumes.

## 3. Materials and methods

In this section we describe the device we used for video recording and gaze estimation, as well as the segmentation methods and task-specific logic for volume and mass estimation. Figure 1 describes the use the gaze and models used for each of the tasks.

### 3.1. Devices

Two devices have been used for our experiments, both from *Pupil Labs*[1]. The *Invisible* device is capable of streaming RGB video at up to 30 frames per second with a resolution of $1080 \times 1080$ pixels and gaze at 120 Hz. The *Neon* device is capable of streaming RGB video at up to 30 frames per second with a resolution of $1200 \times 1600$ pixels and gaze at 200 Hz. Whereas the *Invisible* model comes with glasses already included, the *Neon* model required to 3D-print glasses to hold the recording module.

---

1. http://www.pupil-labs.com

## 3.2. Temporal segmentation

Our approach uses the gaze samples to compute fixation coordinates, and by choosing the gaze that is closest to the fixation coordinates, the corresponding frame is selected as the fixation frame for the video. In order to compute the fixation coordinates we compute a fixation heatmap $f(x, y)$ by superposing Gaussian distributions centered around the gaze coordinates for each frame using $G$ gaze samples with coordinates $(x_i, y_i)$:

$$f(x, y) = \sum_{i=1}^{G} \exp\left(-\left(\frac{(x - x_i)^2}{2\sigma_X^2} + \frac{(y - y_i)^2}{2\sigma_Y^2}\right)\right) \tag{1}$$

where $\sigma_X^2$ is the variance of the coordinates on the X axis and $\sigma_Y^2$ on the Y axis, respectively. The fixation coordinates are computed as the maximum value of the heatmap, near the highest concentration of gaze points:

$$(x_{fix}, y_{fix}) = \arg\max_{(x,y)} f(x, y) \tag{2}$$

and the fixation frame as:

$$F = \arg\min_{i} \sqrt{(x_{fix} - x_i)^2 + (y_{fix} - y_i)^2}, \quad \text{where} \quad i \in [1, G] \tag{3}$$

## 3.3. Spatial segmentation

The spatial segmentation aims at locating the object of interest within the frame. For this, we explore various possibilities of prompting the de-facto standard segmentation model SAM (Kirillov et al., 2023). We evaluate different prompts, namely using the raw gaze on the fixation frame, running the auto-segmentation mode, and gaze-informed sampling and selection of the region of interest.

### 3.3.1. SEGMENTATION STRATEGIES

For segmentation, we consider the ViT-H SAM flavour and use it to predict segmentation masks in two regimes.

**Prompted** The model is provided with one or multiple seed points contained within the region that we are interested in segmenting. After embedding the image, inference is performed to obtain masks of likely objects corresponding to the prompt.

**Auto-segmentation** SAM's Auto-segmentation mode produces an exhaustive segmentation of an image, by prompting with a regular grid. Therefore the result is a label map in which all pixels are assigned to a non-semantic label, defining (possibly connected) regions with the same label. This converts the classic background/foreground segmentation problem into a problem of identifying which of the regions is the one to be chosen as region of interest. An example of auto-segmentation is shown in Figure 2.

## 3.4. Task-specific logic

In this section we describe the additional logic used for solving each of the end tasks: mass estimation through digit recognition on a digital scale display and volume estimation of liquid measurements with a graduated cylinder.

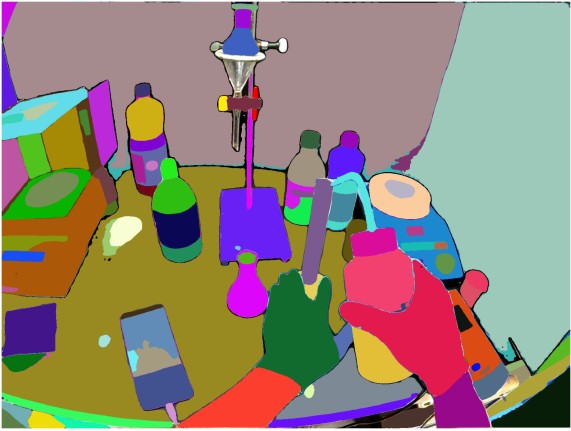

Figure 2: Result of using the Auto-segmentation mode of SAM in a video frame. The image is exhaustively segmented so that all pixels are assigned to a labelled region.

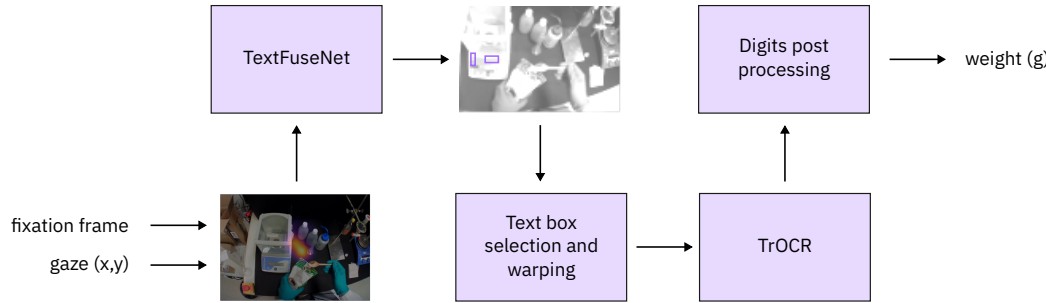

Figure 3: Schematic overview on sample data of the steps involved in the mass estimation process. The gaze-based segmentation algorithm is used to detect the frame of interest and the corresponding fixation, then ext detection region proposals are computed with TextFuseNet. Regions fitting in a horizontal rectangle are retained and warped to correct the perspective. The digit transcription is obtained by the TrOCR model, whose output is postprocessed for error correction.

### 3.4.1. MASS ESTIMATION

The mass esimation pipeline leverages zero-shot inference on state of the art models, namely TextFuseNet for text detection in the wild (Ye et al., 2020) and the transformer-based model for Optical Character Recognition TrOCR (Li et al., 2023a). A schematic overview of the steps involved in the process is presented in Figure 3.

TextFuseNet is a deep learning model with convolutional backbones that operates at three different levels to obtain rich feature representations of the imaging data. Different features are extracted to represent characters, words and global appearance and then fused into a unique, rich representation that facilitates the text detection (Ye et al., 2020). As a first step (after the temporal and optional spatial segmentation of the frames containing

displays in the videos), we retain the TextFuseNet region proposals with a confidence level higher than 0.7. Since we are only interested in the digits displayed by the lab equipment, we expect a text region that can be inscribed in a horizontal rectangle. To avoid focusing on uninformative content such as brand logos and single characters, we only retain the first two region proposals with the largest area for which the region height is larger than the width.

Once the text region proposal is obtained, the next step is the transcription of the digits. At this point, most of the regions identified by TextFuseNet are displaced with an orientation that is not frontal to the camera. This is normal, because the operator is naturally interacting with the environment and is observing the instrument measurements from different angles. To correct the perspective for OCR, we perform basic image processing operations such as image warping. We first find the contours of the region proposal and approximate them into a polygon coordinates. We then use image warping to change the perspective and fit the polygon into a rectangle with matching width and height. The warped text regions are then passed through the TrOCR model, which predicts the transcription into a string.

As a last step, the postprocessing of the OCR predictions ensures that basic mistakes from the OCR model are removed, for example replacing the mislabeling of a digit for a letter. Prior knowledge on the instrument is used here to replace wrongly predicted letters into digit values and to ensure that the predicted mass is reported in grams with a precision compatible with the scales used to perform the measurements.

### 3.4.2. Volume estimation

We leverage SAM (Kirillov et al., 2023) to estimate the masks associated to the graduated cylinder and the liquid contained inside. Figure 4 describes the steps followed to obtain the masks of the container ($M_c$) and liquid ($M_l$). We use the two approaches for segmentation as described in Section 3.3. For *Auto-segmentation*, we leverage the fixation gaze to identify the candidate container masks, by selecting those that include the gaze coordinates. For *Prompted* approach, we directly prompt SAM with the fixation coordinates and output the top three masks produced by the model. After that, for both approaches, we filter the masks and select the biggest one, whose aspect ratio is compatible with the dimensions of the graduated cylinder. To take into account possible object rotations or noisy masks, we define an acceptable ratio between the width and the height of the object to be comprised between $1/5$ and $1/8$. We filter the mask further to remove the base of the graduated cylinder by considering a maximum 10% variation from the median value of the mask's width. The obtained mask corresponds to $M_c$. In the case in which none of the masks satisfy the aspect ratio threshold, we flag the video and do not assign any masks and volume estimation to it. Both segmentation methods involve a second step which requires re-prompting of SAM with the coordinates of a *virtual gaze* to estimate $M_l$. The position of the *virtual gaze* is determined to be on the vertical axis of symmetry of the container mask and height defined as a parameter that was fixed during tuning. We filter out the smallest mask whose width is compatible with $M_c$ among the 3 outputs produced by the model. We then apply equation 4 to estimate the volume based on a $M_c/M_l$ pair that satisfies the aspect ratio and shape requirements. We flag the video as *without prediction* in all the other cases. $M_c$ and $M_l$ can then be used for direct estimation of the liquid volume in milliliters (mL) units through Equation 4.

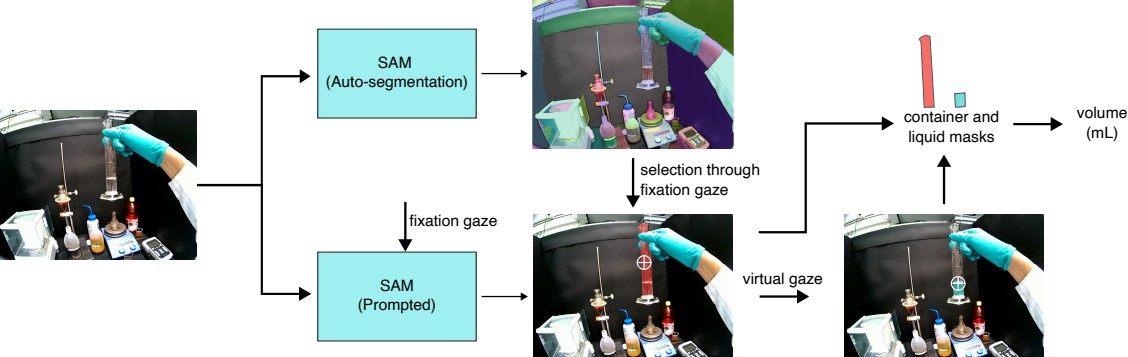

Figure 4: Schematic overview on sample data of the steps involved in the volume estimation process. For each video, a fixation frame is detected. SAM is then used to estimate the container mask based on gaze-prompting. To detect the liquid mask, the model is re-prompted with a virtual gaze coordinate. Liquid volume is estimated directly from detected container and liquid masks.

$$\frac{h(M_l)}{h(M_c)} \cdot K_c \qquad (4)$$

where $h(M_l)$ and $h(M_c)$ are the heights of the liquid and container masks, respectively. The term $K_c$ corresponds to the sum of the nominal cylinder capacity and a correction factor, which takes into account the top part of the object that is not graduated. A standard 50 mL graduated cylinder was used for liquid measurement by the users of the smart goggles in all the laboratory experiments. The measured additional volume head was determined to be 19 mL, resulting in a $K_c$ factor equal to 69 mL. Unless specified differently, this value was fixed for all the experiments and results reported hereafter. The use of the heights of the segmentation masks was preferred over the use of masks' areas to minimize the impact of non-continuous or noisy masks on the volume estimation.

## 4. Experiments

### 4.1. Data sets

A set of 460 videos while measuring liquids and 231 videos while weighting solids have been collected. Because the intention is to use the proposed system in real time or close to real time, the videos were of limited length and frame rate. Each video consists of 16 frames that span 4 seconds. The length of the videos was chosen to be aligned to the duration of the tasks, and the frame rate to be compatible with video transformers. Each video has been manually annotated and tagged to keep track of actions that were completed or incomplete, occlusions or other perturbations, location and operators using the egocentric video device. A summary of these splits is shown in Figures 5 for volume estimation and Figure 6 for mass estimation.

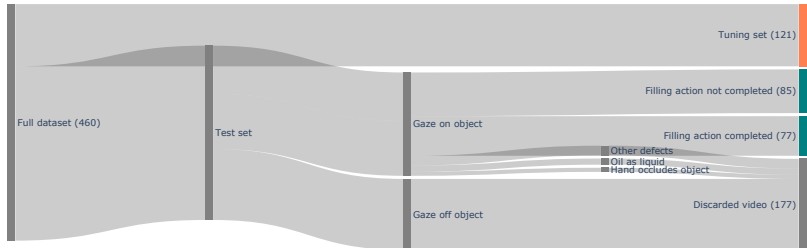

Figure 5: Diagram of the volume estimation dataset splits. From an original dataset of 460 videos, they were split in a subset of videos to tune the parameters of the method and a set of videos kept for test only. Among those, they were manually tagged an annotated until achieving the final splits (highlighted in the diagram).

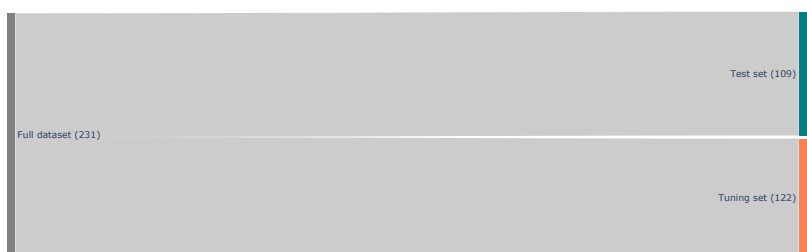

Figure 6: Diagram of the mass estimation dataset splits. From an original dataset of 231 videos, they were split in a subset of videos to tune the parameters of the method and a set of videos kept for test only.

### 4.2. Experimental settings

To investigate the impact of the gaze estimation we explore various configurations for each task settings.

**Mass estimation**  We conduct ablation studies defining different strategies to leverage the gaze estimation: (i) using only temporal segmentation to determine the fixation frame and analyzing the full frame (*Fixation*); (ii) using only spatial segmentation prompted with gaze on a randomly sampled frame (*Crop*); (iii) applying both temporal and spatial segmentation (*Fixation and crop*); (iv) no use of the gaze operating on a full randomly sampled frame (*No gaze*).

**Volume estimation** We conduct ablation studies defining different strategies to leverage the gaze estimation using both the (*Auto-segmentation*) and direct prompting on the fixation frame (*Prompted*) by: (i) using only temporal segmentation (*Temporal*); (ii) selecting a random frame as the fixation frame (*Random Fixation*); (iii) using the center of the fixation frame instead of the actual gaze to generate the container mask (*Frame center*); (iv) in order to correct for calibration errors and to perform robust spatial segmentation, we explore the use of Monte Carlo rejection sampling to obtain a list of points that can be used to prompt SAM (*Temporal prompt sampling*). We draw 20 points from a probability map built using the gaze coordinates $x_F, y_F$, corresponding to the fixation frame (Section 3.2) as shown in Equation 5.

$$p(x, y) = \exp\left(-\left(\frac{(x - x_F)^2}{2\sigma_X^2} + \frac{(y - y_F)^2}{2\sigma_Y^2}\right)\right).$$ (5)

### 4.3. Evaluation and metrics

We evaluated the two end tasks by computing the mean absolute error (MAE). For mass estimation, the MAE exposes directly the deviation from the true value and is measured in grams. Instead, the MAE for the volume estimation is measured in milliliters. In both end tasks, it is possible to have multiple estimations for the same video (for mass estimation this happens when multiple text region proposal are obtained and processed; for volume estimation this happens when we have an estimation for multiple frames in the same video by deactivating the temporal fixation), when this happens we compute the median of the available values related to the same video sample.

#### 4.3.1. BASELINES

In order to benchmark our results, we synthetically define baselines using the measurements observed in the tuning samples. As the videos recorded different executions of an experimental protocol with slight variations where measured quantities are almost constant, we considered the average of the measurements in the respective tuning sets as baselines for both mass and volume pipelines.

#### 4.3.2. SCORING MASK COHERENCE FOR VOLUME ESTIMATION

In tandem with the development of a pipeline for liquid and container segmentation, we have devised a comprehensive scoring mechanism to assess their topological coherence. This score, born out of the need for masks that align with physical constraints, incorporates multiple components based on overlap, spatial positioning, and adherence to container dimensions. The scoring process involves analyzing a pair of masks, i.e., the ones for liquid and container, and assigning a numerical value $s_a \in [0, 1]$ that correlates with the likelihood of matching the respective targets. The scoring mechanism is the result of an aggregation of four distinct metrics. The first metric assesses the mask overlap ($s_i$), defined as the sum of the intersection size of the two masks divided by the sum of the size of the liquid mask. To compute the other three components of the score, we apply a filter to the container mask, which undergoes a noise reduction process through successive applications of "erosion" and "dilation" operations (for both, we consider square kernels of size $20 \times 20$). Next, we

delineate the boundaries of the container region suitable for accommodating the liquid, i.e., removing areas that cannot contain liquid, such as the base. Assuming a uniform occupation on the entire width of the container, we process the liquid mask, eliminating irregular shapes. Subsequently, for both the processed liquid and container masks, we identify the centroid, highest, and lowest points. We then calculate the vertical distance between the lowest points of the liquid and the container. This distance, measured in pixels, ideally should be zero if the liquid's bottom aligns with the cylinder's base. A non-zero value indicates a misplacement of the liquid mask. To standardize this distance, we divide it by half of the container's height, creating an adimensional ratio: $r_{lg}$. To ensure uniformity with other score metrics, we further process this ratio transforming it into score as follows:

$$s_{lg} = 1.0/(1.0 + 20.0 * r_{lg}{}^2) \tag{6}$$

We also incorporate a filling component into the score, calculated as the ratio between the size of the intersection of the processed liquid and container masks and the size of the container mask, yielding $r_f$. Subsequently, we compute the $s_f$ using the formula:

$$s_f = 1.0 - 4.0 * (r_f - 0.5)^2 \tag{7}$$

This score penalizes overly full or completely filled containers, occurrences often attributed to assigning the same mask to both the liquid and container or due to a liquid mask being too large, possibly influenced by background objects in the scene.

The fourth component of the score evaluates the relative position along the vertical direction of the centroids of the liquid and the container. If the centroid of the liquid is below the centroid of the container, the $s_c$ is set to one; otherwise, it's zero. This criterion aligns with the common-sense notion that the liquid cannot be positioned higher than the container. The final score is the product of the four scores:

$$s_a = s_i * s_{lg} * s_f * s_c \tag{8}$$

## 5. Results

### 5.1. Mass estimation

The results of the pipeline for mass estimation are shown in Figure 7, where we use MAE and Levenshtein similarity score to evaluate the pipeline performance. In Subfigure 7(a), we report the MAE for the four strategies in relation the baseline defined by the mean mass measurement from the tuning set. Of the four strategies we tested two are better than the baseline in terms of absolute error, while the best strategy (*Crop*) processes less than half of the samples compared to the *Fixation* strategy leveraging gaze for temporal segmentation. Interestingly, not using the gaze results in a consistent decrease in the performance of the pipeline. The same trend is highlighted in Subfigure 7(b) when looking at the Levenshtein similarity score. Here it is evident how all strategies are above the baseline and the gaze-based ones are superior compared to the *No gaze* one.

### 5.2. Volume estimation

The results on the liquid volume estimation are shown in Figure 8 and compared to the baseline displayed as solid line. In particular, we report the results on two test splits (4.1)

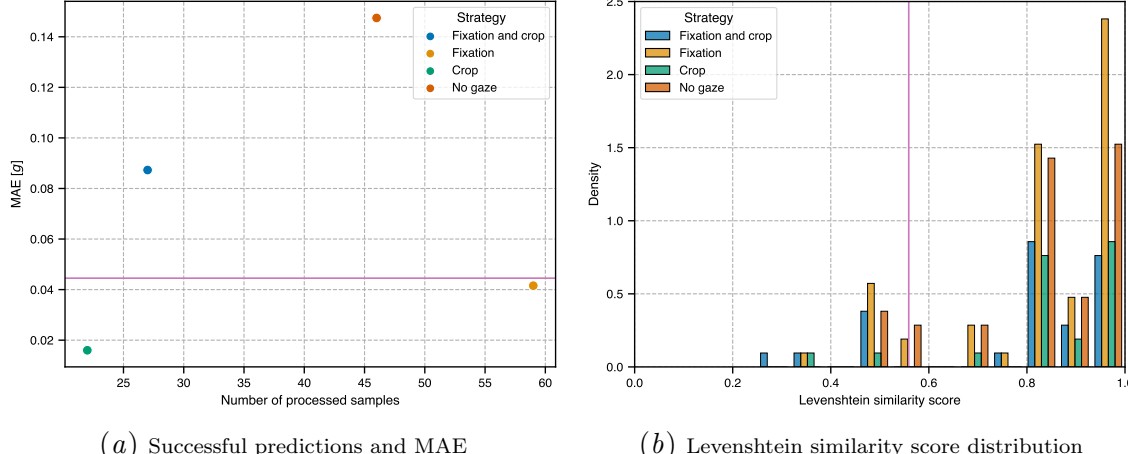

$(a)$ Successful predictions and MAE $\qquad$ $(b)$ Levenshtein similarity score distribution

Figure 7: Evaluation results of mass estimation. Subfigure 7$(a)$ shows that not all strategies produce valid estimations. The MAE for successful predictions is compared to a baseline that predicts the mean value from the tuning set. Subfigure 7$(b)$ shows the distribution of the Levenshtein similarity score for each strategy, compared to a baseline that predicts the mean value from the tuning set.

for *Filling action not completed* and *Filling action completed*, to distinguish the cases in which the graduated cylinder was being filled from those in which the same action was being completed or had already finished. The MAE for prediction of liquid volume is shown in panels 8$(b)$ and 8$(a)$, while the absolute error distributions for the two splits on the *Temporal* fixation setting are reported in 8$(c)$ and 8$(d)$, respectively. In the former two plots, we study the MAE obtained under the *Auto-segmentation* and *Prompted* segmentation strategies and the four fixation settings (Section 4.2). The MAE obtained on the videos that were labeled as *Filling action not completed* is consistently worse than the baseline with error varying in the range $\sim$ 8-17 mL. On the contrary, for the videos that were annotated as *Filling action completed*, the MAE was found to be better than the baseline for the *Prompted* segmentation with *Temporal prompt sampling* and the *Auto-segmentation* approach with *Temporal* and *Temporal prompt sampling*. The latter two cases achieved the best performance with MAE $\sim$ 3.5 mL. The highest number of predictions was obtained for the *Temporal* setting with *Prompted* (26 predictions) and *Temporal prompt sampling* with *Auto-segmentation* (21 predictions). Interestingly, the *Auto-segmentation* approach demonstrates superior performances with respect to the *Prompted* segmentation for all the four different settings. Regarding the error distributions shown in panels 8$(c)$ and 8$(d)$, it can be noticed that the majority of the predictions obtained with the *Auto-segmentation* approach fall in the lower error range $<$ 2.5 mL for both test splits. Instead, the error distribution for the *Prompted* segmentation is more homogeneous and the majority of the predictions falls in the error range that is greater than the baseline predictor. In Figure 9, we report the use of the mask coherence score as a threshold to filter samples. The threshold, ranging from 0 to 1, is applied to filter samples based on the mask coherence score described in equation 8. The MAE is then averaged over samples with scores lower than or equal

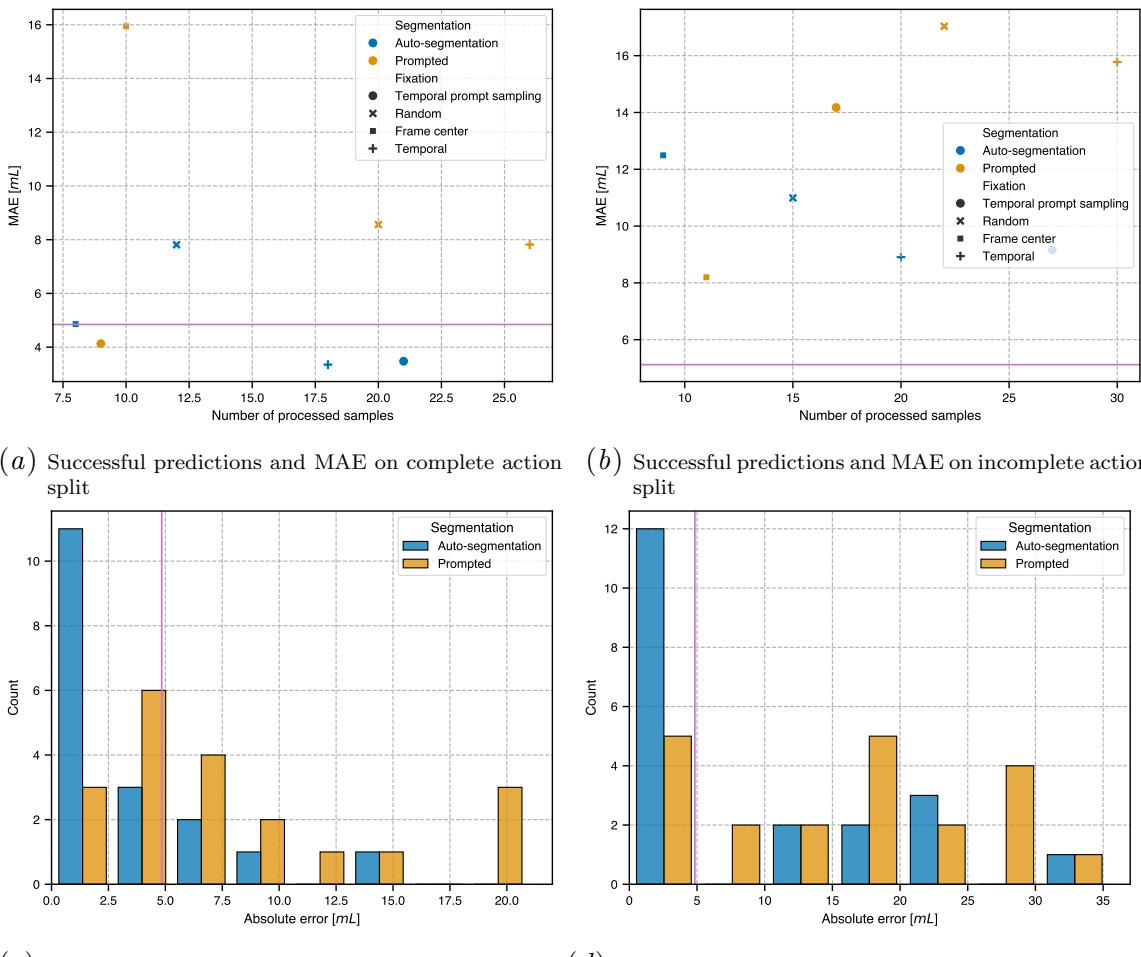

$(a)$ Successful predictions and MAE on complete action split

$(b)$ Successful predictions and MAE on incomplete action split

$(c)$ Absolute error distribution of the best configuration on complete action split

$(d)$ Absolute error distribution of the best configuration on incomplete action split

Figure 8: Evaluation results of volume estimation on the two test splits. Subfigures $8(a)$ and $8(b)$ show that not all configurations can produce a valid estimation, which is visible by a larger MAE than the benchmark for the incomplete action split. Subfigures $8(c)$ and $8(d)$ show the absolute error distribution for the best configuration.

to the specified threshold. Decreasing the threshold indicates samples with lower scores, suggesting less consistency in the masks of the liquid and container.

## 6. Discussion

The results from Section 5 show that incorporating gaze into higher level computer vision task allows to perform zero-shot estimations for mass and volume, with performance metrics impacted by the methodology for data acquisition and the segmentation pipelines we applied. For mass estimation, we showed how relying on state-of-the-art approaches for text

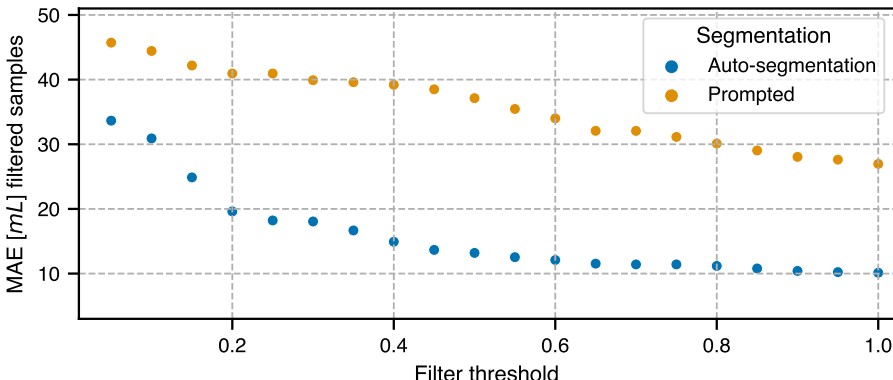

Figure 9: Impact of the mask coherence score to filter out samples. The mean absolute error (MAE) between volume predictions and volume annotations is plotted against various filter thresholds for the auto-segmentation (blue circles) and the prompted segmentation (yellow circles), respectively.

detection and digit recognition in combination with gaze estimates allows us to accurately digitize measurements from egocentric videos with errors in the order of the centigram. Our ablation studies highlight how using the gaze to determine the fixation frame boosts the number of processed samples (*Fixation*), indicating how the extracted frames' quality intrinsically improves when using the gaze. The impact of the gaze is also evident when considering the performance of the *Crop* strategy, where using spatial segmentation driven by the gaze to crop a portion of interest guarantees accurate estimates despite a lower quality of the randomly selected frame (apparent also from the lower number of processed samples). For volume estimation, we found the prediction on videos with incomplete filling actions to be very challenging. We believe that this effect is due to the imperfect estimation of the liquid masks, highly impacted by the liquid flow in the cylinder. For this reason, we foresee the need of an adjunct method that would let the pipeline be applied only to videos in which the filling action has been completed. Instead, the results on the other test set (filling action completed) proves that temporal fixation is effective to enable the identification of the video frame of interest. Indeed, the prediction accuracy and yield are substantially better than the metrics obtained with *Random* fixation frame, regardless of the segmentation approach that was used. Moreover, leveraging gaze in both segmentation pipelines was demonstrated to be extremely beneficial, as the performances drop considerably when the *Frame center* is used as substitute of the gaze prompts. We found that *Auto-segmentation* is generally better than direct gaze-prompting, and that could be explained by the exhaustive identification of all segmentation masks, and thus of the cylinder mask, in a fixation frame. Instead, the latter approach might generate masks with higher noise since the prompt provided to SAM is only the gaze coordinate. Indeed, when using the *Temporal prompt sampling* setting, which provides the model with an ensemble of point prompts, the MAE greatly improves for the *Prompted* approach, although the yield gets reduced. As regards the prediction yield, we find the low prediction rate to be attributed to

the speed of the measurement action, that could result in only a small percentage of frames (out of the 16 available) in which the researcher would actually look at the cylinder or liquid. Thus, the use of spatial and temporal fixation could be limited by videos containing multiple actions. Finally, for our pipeline, we found beneficial to devise an auxiliary metric based on the score described in Equation 8. This auxiliary metric helps identify masks that lack the clear geometrical requirements compatible with both the liquid and the container. Filtering samples with low scores serves as an effective strategy to pinpoint those samples that could lead to substantial errors in predictions. Moreover, this metric relies on a set on rules and it is does not require annotations. As we collected data before the development of the mass and volume estimation pipelines, there are several lessons learned from these experiments. If the data present a minimal spread around a mean value, a normal distribution will perform excellently in such a data set. However, this is not a likely real-life situation. Another lesson learned is that since fixation-based temporal segmentations are effective in reducing the number of frames to analyze and have a positive impact on the absolute error on the end task, estimations over longer videos would allow us to reduce the impact of incomplete actions, which as we have seen are less accurate estimations.

## 7. Conclusion

Herein, we demonstrated that gaze is instrumental in successfully applying state-of-the-art computer vision and foundation modeling approaches to digitize lab measurements. Indeed, accounting for the gaze, we can decrease the absolute error in estimating both mass (digits from a display) and volume of 0.12 g and 4 mL, respectively. These improvements show that using gaze data brings the realization of lab digital twins one step closer, paving the way for unprecedented improvements in the reproducibility of scientific experimentation. Moreover, our contribution highlights how relying on egocentric videos for collecting information in complex environments can primarily benefit from leveraging gaze estimates, defining a new paradigm for data acquisition via vision foundation model-powered pipelines.

### Acknowledgments

The acknowledgments are redacted to preserve anonimity during double-blind peer review.

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
