# OpenReview forum: "Through the eyes of the beholder – a study on zero-shot digitization of lab measurements"
_NeurIPS.cc/2023/Workshop/Gaze_Meets_ML — Submitted to Gaze Meets ML 2023_

### Official Review · Reviewer_maxk · 2023-10-17
**Interesting dataset contribution, weak method contribution**

**Rating:** 2
**Confidence:** 4

**Review:**

The authors propose a method to utilize gaze to detect key video frames and automatically extract liquid volume and object weight from a scale.
The gaze data is used to locate the object of interest form a segmentation mask and then two different pipelines with pre-trained models are used to identify the amount of liquid in a cylinder or the weight shown on a scale.
The authors recorded a new dataset consisting of 460 short video clips recorded in a laboratory with an egocentric camera and gaze tracking.

While the paper demonstrates how gaze could be helpful to automatically log important lab activities, I have strong doubts about the general applicability of their proposed method.
The method contribution seems to be very trivial:
As far as I understand from the limited explanation of their method, for the temporal segmentation they accumulate the raw gaze points into a saliency map and extract a single fixation by taking the location of the maximum value in the saliency map.
Then they select one video frame of interest based on where the gaze point with the smallest distance to the extracted fixation is present.
In other words they identify the fixation with the longest duration and select the video frame that contains the nearest gaze point to that fixation.
I wonder why the authors did not chose to use an established fixation detection algorithm for this.
With their approach it seems to be possible that a random raw gaze point that does not belong to the actual fixation (for example a saccade that intersects with the fixation point) is closest to the fixation and thus, an incorrect frame of interest would be selected.
Their segmentation method seems also to be very trivial in that they simply use an of-the-shelve segmentation model and pick the segment where the selected fixation landed on.

The authors motivated their work by arguing that long data streams are difficult to analyze, log and store, but then use a dataset containing only a single action in a four second clip. If the video clip would contain multiple actions, their method would only be able to log one action since they only consider one fixation from the whole clip.
Furthermore, as shown in Figure 5, the authors did not consider video clips where the gaze data is not on the target object (cylinder, scale), which are therefore not part of the evaluation but are clear cases where the proposed method would fail.
Another concern I have is that the authors defined multiple rules by hand to validate detected segments such as the ration of width and height of the segments.
These rules are likely to lead to incorrect results in other settings where a different scale or different cylinders are used.

While the discussed topic in general seems relevant, the method proposed by the authors does not seem to be in any way robust and only works in this very specifically constructed toy example.

---

### Official Review · Reviewer_pB32 · 2023-10-23
**A study joins several vision models and incorporates gaze to assist lab measurements in egocentric videos but lacks scientific novelty**

**Rating:** 4
**Confidence:** 4

**Review:**

This paper evaluates the use of gaze estimation in egocentric video recordings to infer the associated numerical values of two human actions in a laboratory setting: volume estimation of a liquid measurement and mass estimation of solids on a scale. The authors proposes that the synergies between gaze estimation and computer vision can reduce the burden in some contexts. The study investigates the application of gaze estimation to prompt dual segmentation, using temporal segmentation to identify the frame(s) of interest and spatial segmentation to identify the object the user is looking at.

There are two main parts in the paper. The first one  comprises a method for temporal segmentation using gaze samples to compute fixation coordinates and spatial segmentation using the standard segmentation model SAM. The method uses the ViT-H SAM flavour to predict segmentation masks and auto-segmentation mode to produce an exhaustive segmentation of an image. The text also discusses task-specific logic for mass estimation through digit recognition on a digital scale display and volume estimation of liquid measurements with a graduated cylinder. The method leverages zero-shot inference on state-of-the-art models like TextFuseNet for text detection.

The second part is a method for converting text regions into digits using TrOCR, and postprocessing to remove errors. The method also uses SAM to estimate masks for a graduated cylinder and its contents. The process involves filtering the masks and determining the aspect ratio for the cylinder. The volume is then estimated using equations, and the volume is then calculated using the cylinder's nominal capacity and a correction factor.

The authors presented a related work on the video scene segmentation, however, they did not explicitly mention which algorithm they used for this purpose. If video scene segmentation is not performed, then the temporal segmentation step explained in Section 3.2 would not work correctly since the calculated heatmaps' extremums might be obtained from gaze points that actually fall on different world scene locations and would lead to incorrect results.

The method works in a highly constrained and controlled lab environment. This caused to accept many priors such as the volume of cylinders needs to be known or digital screens of lab scales' to have the largest area in the image among all other text information. Besides such assumptions, the method also uses too many rules to extract the desired outcomes which calls into question the robustness of the method. For instance, the aspect ratio of the cylinder needs to be in between 1/5 and 1/8 to be detected properly.

Briefly, the study relies on too many assumptions that work through the information extracted by several deep learning architectures, i.e., SAM, TextFuseNet and TrOCR; and then the extracted information is further processed and filtered by using the gaze data. In other words, gaze data was not directly involved in a machine learning routine in the study; instead it is used to interpret the results obtained from other architectures. For instance, in the volume estimation task, fixation points are employed to select the graduated cylinder among many other objects segmented by SAM in the image. This significantly reduces the novelty of the study. I would expect those assumptions or rules would be avoided and replaced with a novel machine learning/computer vision approach that jointly runs along with the other models in an end-to-end fashion.

In conclusion, paper presents a study that successfully joins several vision models and used gaze data as a simple input with lots of assumptions. In its current style, it might be a nice application paper that implements a working system for interpretation of egocentric videos in a highly constrained lab environment, but it lacks novelty in terms of scientific contribution.

Minor issues:
pg. 2: In this paper, we evaluate the use "OF" gaze estimation in egocentric video recordings...

pg. 6 (caption of Fig. 3): ... then "T"ext detection region proposals are computed with TextFuseNet.

---

### Official Review · Reviewer_j55m · 2023-10-24
**The idea is imaginative but the presentation often lacks clarity and omits important details**

**Rating:** 5
**Confidence:** 3

**Review:**

The authors present a methodology for using gaze prompting combined with computer vision to automatically track lab measurements. They also collected their own dataset of 460 videos to develop their approach. While the idea has merit and is imaginative, the manuscript is confusing and hard to understand at times. If the authors improved the grammar and clarity of the manuscript, it could significantly improve the value of their contribution.

**General comments**

- The idea is imaginative
- Clarity needs to be improved, a lot of sentences are not very elegantly written or are hard to understand: “…where tasks can hardly be interrupted to annotate details”, “collecting large scale collections of video recordings”, “the impossibility to stop the action, safety measures, the need of immediacy or human inattention”
- You have a tuning set to “tune the parameters of the method”, but I'm not sure what you mean or what these parameters are
- Grammar needs improving: ”because of the surgeon’s impossibility of stopping the procedure, and this is known to cause unpredictability in the outcomes”
- The videos are very short, it seems like in a realistic setting you would have way more noise. It would be interesting to see the performance in a real-time and real-life scenario
- How did you choose the 16 frames and 4 seconds you use as input?

**Specific comments**

- “such as smartphones, smartwatches, and smart googles”: did you mean smart glasses?
- “Because the intention is to use the proposed system in real time or close to real time, the videos were of limited length and frame rate.”: Reducing the size of the video does not mean your approach works in real time. Choosing an efficient model does.
- “Each video consists of 16 frames that span 4 seconds. The length of the videos was chosen to be aligned to the duration of the tasks, and the frame rate to be compatible with video transformers.”: do you mean the number of frames? there are various sizes of video transformers with different input length expectations. What do you mean the length is aligned to the duration of tasks?
- Figures 5 and 6 seem unnecessary, you can add this information in the text
- I don't understand how you use the Levenshtein similarity score or what Figure 7b is showing and I see no explanation in the text of either of those
- Why show Figure 8c and 8d?

---

### Meta-Review · Area_Chair_MfLN · 2023-10-26

**Recommendation:** Reject
**Confidence:** 4

**Metareview:**

Reviewers have raised concerns regarding the paper's clarity and its perceived need for more scientific novelty and method robustness. Additionally, they've highlighted the need for enhanced grammatical construction and paper clarity. Furthermore, the reviewers have requested more details on design choices, precisely the rationale behind selecting 16 frames and 4-second videos as input. I recommend addressing these concerns and enhancing the paper accordingly.

---

### Decision · Program_Chairs · 2023-10-26

Reject